# Music Therapy Is Effective during Sleep in Preterm Infants

**DOI:** 10.3390/ijerph18168245

**Published:** 2021-08-04

**Authors:** Susann Kobus, Marlis Diezel, Monia Vanessa Dewan, Britta Huening, Anne-Kathrin Dathe, Ursula Felderhoff-Mueser, Nora Bruns

**Affiliations:** Clinic for Pediatrics I, Essen University Hospital, University of Duisburg-Essen, 45147 Essen, Germany; marlis.diezel@gmail.com (M.D.); Monia.Dewan@uk-essen.de (M.V.D.); Britta.Huening@uk-essen.de (B.H.); Anne-Kathrin.Dathe@uk-essen.de (A.-K.D.); Ursula.Felderhoff@uk-essen.de (U.F.-M.); nora.bruns@uk-essen.de (N.B.)

**Keywords:** music therapy, preterm infants, sleep, neonatal intensive care unit, premature infants

## Abstract

Recent research found evidence supporting music therapy for preterm infants to stabilize vital signs and possibly promote neurodevelopment. Even though preterm infants spend a considerable amount of time sleeping, the effectiveness of music therapy during sleep has not been studied. The aim of this study was to investigate the effect of music therapy on preterm infants’ vital signs with respect to the state of wakefulness. The first 20 consecutive infants born with <32 weeks’ gestational age (GA) from the intervention group of an ongoing randomized controlled trial received live music therapy twice a week until hospital discharge. The heart rate, respiratory rate, oxygen saturation, and state of wakefulness were recorded before and after therapy. We observed significantly lower heart and respiratory rates and higher oxygen saturation after live music therapy sessions in general (mean differences −4.9 beats per min; −7.0 breaths per min and +1.5%, respectively). When music therapy was applied during sleep, respiratory rates significantly lowered by 8.8 breaths per min and oxygen saturation increased by 1.6%, whereas in the awake state the vital parameters did not significantly change (heart rate −5.2 beats per min; respiratory rate +0.6 breaths per min and oxygen saturation +1.0%). Music therapy stabilized the respiratory rates and oxygen saturations in sleeping preterm infants.

## 1. Introduction

From a gestational age of 26 weeks, the human fetus can perceive auditory stimuli [1]. Maturation of the auditory system is influenced by the acoustic environment of an infant and is the basis for later language development, learning and memory formation [2]. In infants born prematurely, this maturational process is disrupted. Preterm birth coincides with a period of rapid brain development, making the infant’s brain highly vulnerable but responsive at the same time to neuroprotective interventions [1].

A promising family-integrated early intervention to improve infant development, parental well-being, and bonding is music therapy [3]. It is feasible and well-tolerated by infants, parents, and staff even after extremely preterm birth [4,5]. Parents and staff perceived music therapy as valuable for development-promoting care in the neonatal intensive care unit [5]. Infants were not overstimulated by live-performed music therapy [5]. Furthermore, music therapy attenuates infants’ stress in the neonatal intensive care unit, even though the strength and sustainability of the effect remains unclear [6]. Several studies showed that music therapy had a stabilizing and relaxing effect on the preterm infant’s general behavioral status, sleep patterns, and vital signs [7,8,9,10,11]. Live music and live sung lullabies were superior compared to recorded music in reducing the heart rate and improving behavioral scores [12]. 

Even though newborns and preterm infants spend approximately two thirds of the daytime sleeping during their first weeks of life [13], we found no study that assessed the effect of music therapy during sleep compared to wakefulness. From our own experience, some staff members and parents raise concerns that music therapy is not effective when applied during sleep. The aim of this study was to assess if live music therapy affected preterm infants’ vital signs when applied during sleep.

## 2. Materials and Methods

### 2.1. Study Design

The study was designed as a prospective, randomized controlled clinical trial which is still ongoing. Infants were randomly assigned to either music therapy (intervention group) or no music therapy (control group). There were no differences between the groups in medical care. To investigate the effects of music therapy on vital signs we analyzed the protocols of 307 music therapy sessions of the first 20 consecutive infants that had been allocated to the intervention group.

### 2.2. Eligibility and Recruitment

Infants born before 32 weeks’ gestation at the University Hospital Essen between October 2018 and November 2019 were eligible for the study. Because the study was designed to examine the neurodevelopment outcome later in the course, exclusion criteria were congenital hearing disorders, periventricular haemorrhagic infarction, cerebral malformations, and underlying diseases that impair neurological development. 

Parental informed consent was obtained during the first week of life, at a minimum age of 72 h. The study was approved by the local ethics committee of the Medical Faculty of the University of Duisburg-Essen (18-8035-BO).

### 2.3. Intervention

Music therapy was performed twice weekly in clinically stable patients from the second week of life until discharge. The timing of each therapy session was coordinated by the music therapist, nursing staff and parents. During music therapy, the infant remained in the same position as before therapy, e.g., the incubator, heated cod or parent’s arm or breast during kangaroo care.

Each session consisted of individual, improvised singing by the music therapist and/or the use of the instrument sansula. Improvised singing was guided by the infant’s breathing and reactions, beginning with humming tones that were followed by tone sequences with improvised text. The sansula consists of a wooden ring covered with an eardrum, on which a small kalimba is attached. It creates a space-filling, long-lasting and soft sound. 

Music therapy was carried out for each child individually at a low volume and could not be perceived by other patients treated in the intensive care unit. Vital signs (heart rate, respiratory rate, and oxygen saturation), wakefulness, and physical contact during the session were documented in a protocol before and after each therapy session. Clinical data were retrieved from the patients’ medical record.

### 2.4. Statistical Analysis

Quantitative variables are presented as the mean and 95% confidence intervals or standard deviation, respectively; for qualitative factors, absolute and relative frequencies are given. Therapy sessions were stratified depending on the corrected gestational age at intervention into the age groups 24 to 27 weeks, 28 to 31 weeks, 32 to 35 weeks, 36 to 39 weeks and 40 to 43 weeks to assess the plausibility of our data. Depending on the infants’ wakefulness before and after therapy, we classified infants into four categories (stayed asleep, fell asleep, woke up and stayed awake). 

The effect sizes for pre- and post-therapy vital sign values were calculated as suggested by Maier–Riehle for single group pre-post study designs [14]. All statistical calculations were performed using SAS Enterprise Guide 7.1. (SAS Institute Inc., Cary, NC, USA). *p* values < 0.05 were considered statistically significant. 

## 3. Results

### 3.1. Patients

Seventy-eight infants at <32 weeks’ gestational age were treated at the University Hospital Essen during the study period. Forty infants were included into the main study, 20 in the therapy group and 20 in the control group. The clinical characteristics of the included patients in the therapy and the control group are presented in Table 1.

Thirty-eight infants were not included. The reasons for exclusion were birth in another hospital (*n* = 3; 8%), death before recruitment (*n* = 8; 21%), transfer to another hospital (*n* = 2; 6%), periventricular hemorrhagic infarction (*n* = 1; 3%), maternal critical illness (*n* = 2; 6%), insufficient German language skills to understand objectives of the study (*n* = 3; 8%), and lack of interest to participate (*n* = 18; 47%). Excluded infants were born at a mean gestational age of 28 + 0 weeks (range 22 + 4 to 31 + 3 weeks) with a mean weight of 1056 ± 467 g (range 210 to 1860 g). 

### 3.2. Music Therapy Sessions

307 music therapy sessions were conducted in the therapy group between gestational ages of 24 + 0 and 43 + 0 weeks (Table 2). Forty-seven sessions were conducted with the infant awake and 150 during sleep. The mean duration of each music therapy session was 19.3 ± 4.3 min (range 10 to 50 min).

### 3.3. Analyses by Gestational Age

#### 3.3.1. Heart Rate

We found an overall decrease in the heart rate after therapy of 4.9 beats per min (95% CI 6.6–3.3) (Table 2). The baseline heart rate before therapy decreased with an increasing corrected GA (Figure 1a). Significant decreases could be observed in infants >32 weeks’ corrected GA (Table 2 and Figure 1a). Accordingly, the calculated effect sizes increased with the GA.

#### 3.3.2. Respiratory Rate

The overall decrease in respiratory rates was 7.0 breaths per min (95% CI 9.2–4.8) (Table 2). The baseline respiratory rate before therapy decreased with an increasing GA but rose again at >40 weeks corrected GA (Figure 1b). Of the 15 therapy sessions at >40 weeks corrected GA, five (33%) were performed in an infant with bronchopulmonary dysplasia. 

Subanalyses showed larger absolute differences and stronger calculated effect sizes at lower corrected GAs (Table 2 and Figure 1b). 

#### 3.3.3. Oxygen Saturation

We found an increase of 1.5% (95% CI 1.0–2.0) for all infants when comparing values before and after therapy (Table 2). The baseline oxygen saturation increased with a corrected GA until term equivalent age (Figure 1c). Stratification showed that infants at >28 weeks’ corrected GA had higher SaO_2_ after therapy (Table 2 and Figure 1c). In infants at >40 weeks, the baseline and post-therapy SaO_2_ were lower. The calculated effect sizes increased with the GA.

### 3.4. Analyses by State of Wakefulness 

Out of 307 sessions, 150 (49%) were performed while infants were asleep. During 79 (26%) sessions infants fell asleep during music therapy, during 31 (10%) they woke up, and during 47 (15%) they stayed awake. The corrected gestational ages did not differ significantly between states of wakefulness (Table 3).

#### 3.4.1. Heart Rate

Infants who fell asleep showed a decrease in heart rates of 14.2 beats per min (95% CI 17.5–10.9) with non-overlapping confidence intervals for baseline and post-therapy values. The changes in the heart rates in the other groups were less pronounced (Table 3, Figure 2a). The calculated effect sizes were strongest in infants who fell asleep (−1.00) and who stayed awake (−0.30).

#### 3.4.2. Respiratory Rate

Infants who stayed asleep showed a decrease of the respiratory rate by 8.8 breaths per min (95% CI 11.7–5.9) and infants who woke up a decrease by 14.1 breaths per min (21.9–6.3) during music therapy. The CIs for the baseline and post-therapy values in both groups did not overlap (Table 3, Figure 2b). The effect sizes were strongest during sleep (−0.57) and in infants who woke up during therapy (−0.84).

#### 3.4.3. Oxygen Saturation

Oxygen saturation increased during music therapy in infants who stayed asleep (1.6% (95% CI 0.9–2.3%)) and who fell asleep (2.3% (95% CI 1.2–3.3%)) (Table 3, Figure 2c). Accordingly, the effect sizes were largest in these two groups (0.31 and 0.48, respectively). 

## 4. Discussion

This study provides the first evidence that music therapy has beneficial effects on vital signs when delivered to sleeping preterm infants. The general changes in vital signs were in line with previously published data, thus enabling stratified analyses [7,11]. We found a decrease in the respiratory rate and an increase in oxygen saturation in infants who were asleep before and after the session. Our data show that music therapy has effects on vital signs when delivered to preterm infants during sleep. While it had previously been shown that music therapy improved sleep patterns in preterm infants [11], the efficacy of music therapy had not been studied with respect to the state of wakefulness. The data from this study suggest that live music therapy can be offered to sleeping infants. This possibility largely facilitates the planning of therapy sessions.

The results of this study support previous findings on the short-term stabilization of the vital signs of preterm infants by music therapy and add new evidence that vital signs stabilize during the intervention even in sleeping infants. However, a transient stabilization of vital signs is not the main purpose of interventions in the neonatal intensive care unit. Interventions that improve the neurodevelopmental outcome at later ages and parental well-being are more important. Music therapy could have effects on the infant or parents that are not reflected by vital signs, e.g., the long-term modulation of stress levels. Parental stress and the anxiety of preterm infants receiving music therapy were lower compared to controls [15]. In line with this, former preterm infants’ fear reactivity at 12 months of age and anger reactivity at 24 months of age were more similar to term-born controls after receiving music therapy in the neonatal intensive care unit compared to preterm controls who had not received music therapy [16]. Magnetic resonance imaging showed that live music therapy promoted functional brain activity and connectivity in preterm infants [17]. These studies give the first evidence that the short-term stabilization of vital signs that we and others observed might translate into a positive long-term development. 

Our study has several limitations. Vital signs were not documented for the control group that received no therapy. An analysis of spontaneous changes in vital signs occurring without music therapy and during different states of wakefulness would have made exact effect size calculations possible. Music therapy sessions were performed at different times of the day and had no standardized temporal distance to feeding, causing a certain inhomogeneity of the recorded data. We did not record data on pre- and post-therapy oxygen supplementation. In our neonatal intensive care unit, oxygen supplementation is guided according to predefined target ranges of oxygen saturation depending on the corrected gestational age of an infant. Therefore, changes in oxygen saturation during the session may have been attenuated or exaggerated due to an adjustment of the oxygen supplementation. Due to the small sample size, subgroup analyses, e.g., on the interindividual response to music therapy at a given gestational age but at different corresponding postnatal ages, could not be carried out.

In conclusion, our study confirms former findings on the stabilizing effect of live music therapy on vital signs. It adds new evidence that music therapy is effective during sleep in preterm infants, making the planning of therapy sessions easier for music therapists, nurses, and parents.

## Figures and Tables

**Figure 1 ijerph-18-08245-f001:**
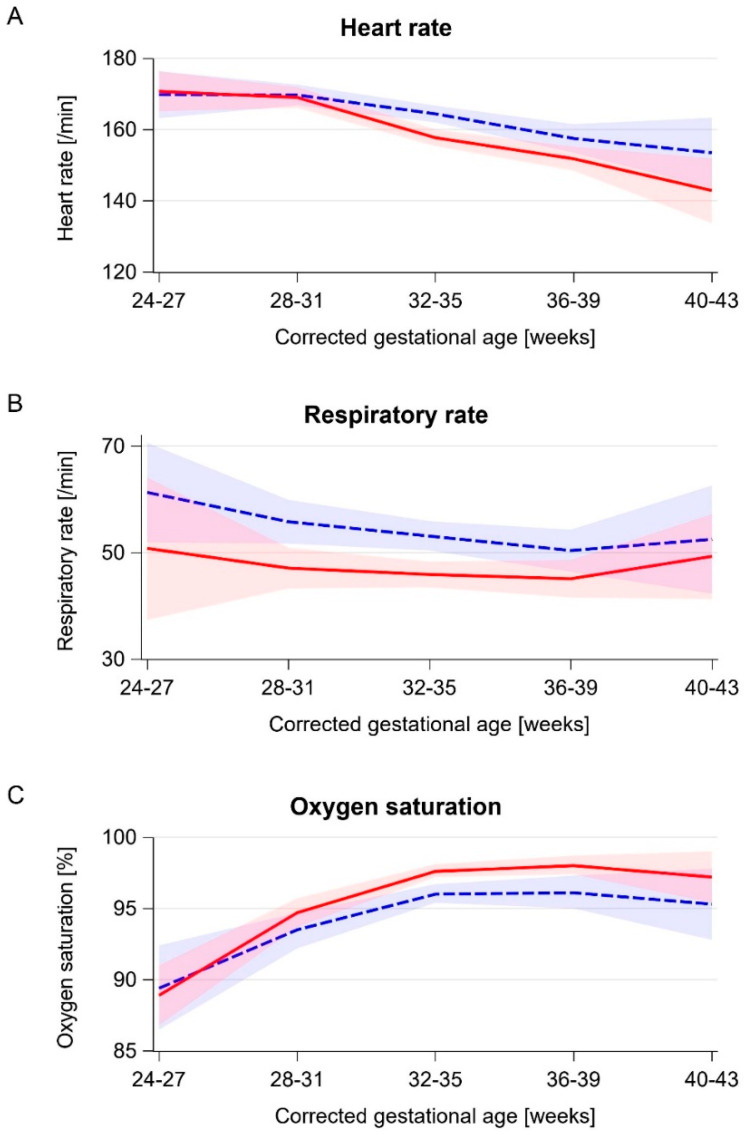
Baseline and post-therapy vital signs by corrected gestational age. Blue dashed line: mean baseline value, red solid line: mean post-therapy value, blue and red bands: corresponding 95% confidence intervals. (**A**) Heart rate. (**B**) Respiratory rate. (**C**) Oxygen saturation.

**Figure 2 ijerph-18-08245-f002:**
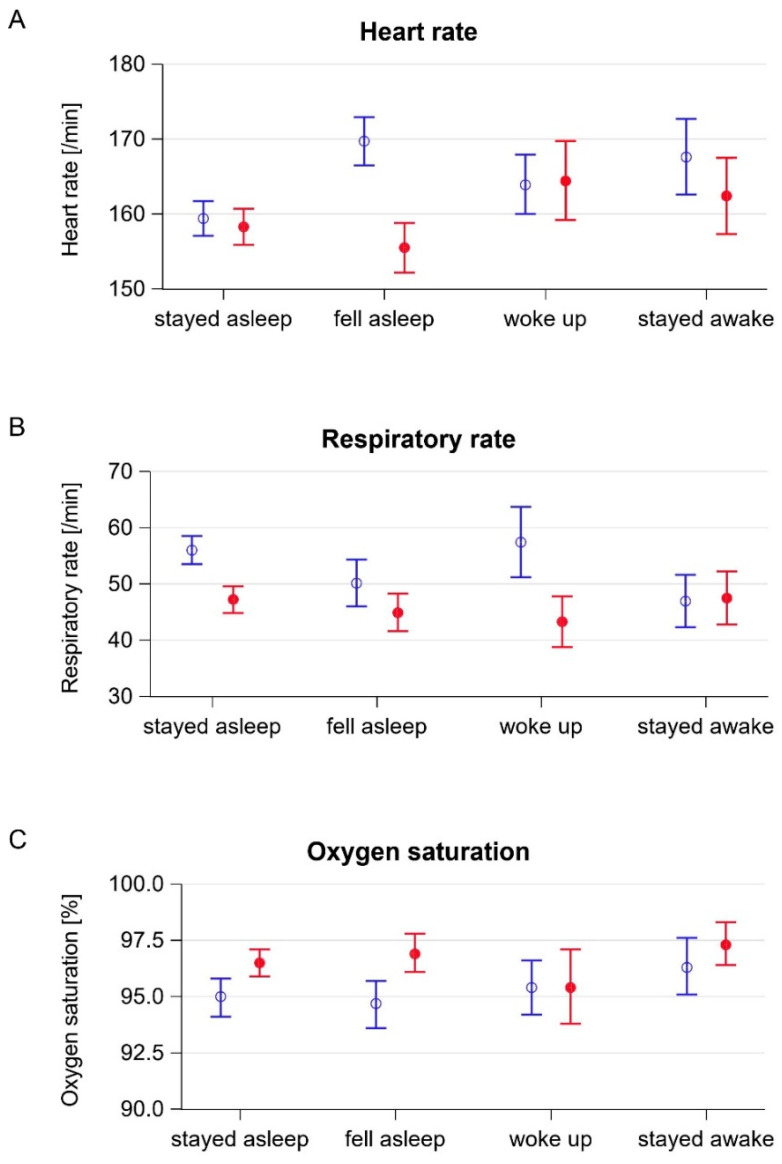
Baseline and post-therapy vital signs by states of wakefulness. Blue empty circle: mean baseline value, red filled circle: mean post-therapy value, blue and red error bars: corresponding 95% confidence intervals. (**A**) Heart rate. (**B**) Respiratory rate. (**C**) Oxygen saturation.

**Table 1 ijerph-18-08245-t001:** Clinical characteristics of the included patients in the therapy and the control group.

	Therapy Group (*n* = 20)	Control Group (*n* = 20)
Male, *n*	12 (60%)	8 (40%)
Gestattional age (weeks) Gestation age (weeks), range	28.6 (±2.4) 23 + 6–31 + 4	28.9 (2.7) 23 + 2–31 + 6
Birth weight (g) Birth weight (g), range	1059 (±389) 340–1700	1201 (±404) 580–2120
Died, *n* (age at death)	1 (9 days)	1 (152 days)
APGAR score at 10 min APGAR score at min, range	8.7 (±0.7) 7.0–10.0	8.7 (±1.0) 6.0–10.0
Early onset sepsis, *n*	5 (25%)	4 (20%)
Late onset sepsis, *n*	4 (20%)	5 (25%)
Bronchopulmonary dypslasia (mild), *n*	1 (5%)	2 (10%)
Bronchopulmonary dypslasia (severe), *n*	1 (5%)	1 (5%)
Intraventricular hemorrhage °I–II, *n*	4 (20%)	3 (15%)
Intraventricular hemorrhage °III, *n*	1 (5%)	0
Necrotizing enterocolitis, *n*	1 (5%)	0
Patent ductus arteriosus (total), *n*	11 (55%)	7 (35%)
Medical therapy, *n*	10 (50%)	7 (35%)
Surgery, *n*	1 (5%)	0
Antibiotic treatment, days	8.7 (10.8)	8.4 (17.7)

Notes. SD = standard deviation. APGAR = Appearance, Pulse, Grimace, Activity, and Respiration. Data are presented as the mean (standard deviation) if not indicated otherwise.

**Table 2 ijerph-18-08245-t002:** Heart rate, respiratory rate and oxygen saturation during music therapy sessions.

Vital Sign	Corrected Gestational Age, Weeks	Sessions, *n*	Mean before Therapy (95% CI)	Mean after Therapy (95% CI)	Mean Difference (95% CI)	Effect Size
Heart rate (beats per min)	all	307	**163.7 (162.1–165.4)**	**158.8 (157.1–160.5)**	−4.9 (−6.6–(−3.3))	−0.33
	24–27	14	169.8 (163.2–176.4)	170.7 (165.1–176.3)	0.9 (−4.8–6.6)	0.08
	28–31	70	169.7 (166.8–172.6)	169.0 (166.1–171.9)	−0.7 (−4.0–2.6)	−0.06
	32–35	139	**164.4 (162.0–166.7)**	**157.7 (155.4–160.0)**	−6.7 (−9.0–(−4.4))	−0.48
	36–39	69	157.5 (153.5–161.5)	151.8 (148.5–155.2)	−5.6 (−9.5–(−1.8))	−0.34
	40–43	15	153.5 (143.6–163.3)	142.9 (133.8–151.9)	−10.6 (−23.3–2.1)	0.60
Respiratory rate,	all	300	**53.3 (51.4–55.2)**	**46.3 (44.6–47.9)**	−7.0 (−9.2–(−4.8))	−0.42
(breaths per min)	24–27	8	61.3 (51.9–70.6)	50.8 (37.4–64.1)	−10.5 (−21.4–0.4)	−0.94
	28–31	70	**55.8 (51.7–59.9)**	**47.1 (43.3–50.9)**	−8.7 (−14.0–(−3.3))	−0.51
	32–35	139	**53.1 (50.4–55.9)**	**45.9 (43.5–48.3)**	−7.3 (−10.3–(−4.2))	−0.44
	36–39	68	50.4 (46.4–54.3)	45.1 (41.6–48.6)	−5.3 (−10.0–(−0.6))	−0.32
	40–43	15	52.5 (42.3–62.6)	49.3 (41.3–57.2)	−3.2 (−14.6–8.2)	−0.17
SaO_2_ (%)	all	306	**95.1 (94.6–95.7)**	**96.6 (96.2–97.1)**	1.5 (1.0–2.0)	0.31
	24–27	14	89.4 (86.5–92.4)	88.9 (86.8–91.0)	−0.5 (−3.8–2.8)	−0.10
	28–31	70	93.5 (92.2–94.7)	94.7 (93.7–95.7)	1.2 (−0.2–2.6)	0.23
	32–35	139	**96.0 (95.4–96.7)**	**97.6 (97.2–98.1)**	1.6 (1.0–2.2)	0.41
	36–39	68	**96.1 (95.0–97.3)**	**98.0 (97.4–98.7)**	1.9 (1.0–2.8)	0.40
	40–43	15	95.3 (92.8–97.8)	97.2 (95.4–99.0)	1.9 (−0.6–4.4)	0.42

Notes. CI = confidence interval. SaO_2_ = oxygen saturation.

**Table 3 ijerph-18-08245-t003:** Response to music therapy by evolution of wakefulness during music therapy sessions.

Vital Sign	Wakefulness	Sessions (*n*)	Mean Corrected Gestational Age (95% CI)	Mean before Therapy (95% CI)	Mean after Therapy (95% CI)	Mean Difference (95% CI)	Effect Size
Heart rate (beats per min)	stayed asleep	150	33.6 (33.1–34.2)	159.4 (157.1–161.7)	158.3 (155.9–160.7)	−1.1 (−3.0–0.8)	−0.08
	fell asleep	79	34.3 (33.5–35.1)	**169.7 (166.5–172.9)**	**155.5 (152.2–158.8)**	−14.2 (−17.5–(−10.9))	−1.00
	woke up	31	33.0 (31.6–34.4)	163.9 (160.0–167.9)	164.4 (159.1–169.7)	0.5 (−5.7–6.6)	0.05
	stayed awake	47	35.0 (33.9–36.2)	167.6 (162.6–172.7)	162.4 (157.3–167.5)	−5.2 (−9.9–(−5.2))	−0.30
Respiratory rate	stayed asleep	149	33.6 (33.1–34.2)	**56.0 (53.5–58.5)**	**47.2 (44.8–49.6)**	−8.8 (−11.7–(−5.9))	−0.57
(breaths per min)	fell asleep	76	34.3 (33.5–35.1)	50.1 (46.0–54.3)	44.9 (41.6–48.3)	−5.2 (−10.2–(−0.3))	−0.28
	woke up	30	33.0 (31.6–34.4)	**57.4 (51.2–63.7)**	**43.3 (38.8–47.8)**	−14.1 (−21.9–(−6.3))	−0.84
	stayed awake	45	35.0 (33.9–36.2)	46.9 (42.3–51.6)	47.5 (42.8–52.2)	0.6 (−4.3–0.6)	0.04
SaO_2_ (%)	stayed asleep	149	33.6 (33.1–34.2)	**95.0 (94.1–95.8)**	**96.5 (95.9–97.1)**	1.6 (0.9–2.3)	0.31
	fell asleep	79	34.3 (33.5–35.1)	**94.7 (93.6–95.7)**	**96.9 (96.1–97.8)**	2.3 (1.2–3.3)	0.48
	woke up	31	33.0 (31.6–34.4)	95.4 (94.2–96.6)	95.4 (93.8–97.1)	0 (−1.7–1.8)	0.00
	stayed awake	47	35.0 (33.9–36.2)	96.3 (95.1–97.6)	97.3 (96.4–98.3)	1.0 (0.1–2.0)	0.23

Notes. CI = confidence interval. SaO_2_ = oxygen saturation.

## Data Availability

Original data will be made available to any qualified researcher upon request.

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
