# Peer review of "Music Therapy Is Effective during Sleep in Preterm Infants"

_ijerph, 2021, doi:10.3390/ijerph18168245_

Round 1

Reviewer 1 Report

The authors of the manuscript entitled "Music therapy is effective during sleep in preterm infants" present vital sign data from the first 20 infants recruited into the intervention arm of a randomised trial assessing the effect of music therapy on neurodevelopmental outcomes in infants born preterm. They found that music therapy decreases heart rate and increases oxygen saturation, particularly with increasing corrected age and during sleep. This is a well-described small study, and though it is limited by the lack of data from the corresponding control group, that is acknowledged upfront. A few minor changes to the way the data are presented would benefit the final manuscript.

Minor comments

  1. I would be interested to know whether there is an effect of gestational age on the effect of the therapy at a give corrected age. For instance, does an infant born at 24 weeks who has spent 6 weeks in the NICU respond in the same way as an infant born at 28 weeks who has been in the NICU for 2 weeks, despite both of them having the same corrected age? There is probably not enough data to answer this, but that may be considered as an additional limitation.
  2. The levels of vigilance are described twice in the methods - in both the intervention and statistical analysis sections.
  3. Table 1 has inconsistent use of square brackets vs parentheses (e.g. for range of antibiotic treatment in the intervention group) and dots vs commas to show fractional numbers (dots used for SDs of GS but commas used for average antibiotic days).
  4. There appears to be a typo on page 3 lines 113-114 regarding the average music therapy session length. Some brackets or parentheses may be missing.
  5. Heart rate could perhaps be presented in the units beats per minute rather than just /min.
  6. For the large tables with lots of confidence intervals like Table 3, it would be useful to bold or somehow highlight the CIs that do not cross 1.

Author Response

Dear Reviewer,

we highly appreciate your critical input and the opportunity to improve our manuscript entitled "Music therapy is effective during sleep in preterm infants". Our changes to the original manuscript are marked using the “Track Changes” function.

All comments were addressed in a point-to-point fashion below:

Comments from Peer-Reviewer:

COMMENT 1

I would be interested to know whether there is an effect of gestational age on the effect of the therapy at a give corrected age. For instance, does an infant born at 24 weeks who has spent 6 weeks in the NICU respond in the same way as an infant born at 28 weeks who has been in the NICU for 2 weeks, despite both of them having the same corrected age? There is probably not enough data to answer this, but that may be considered as an additional limitation.

That is actually an interesting point and during the statistical analyses we discussed if it was possible to investigate this question or similar one. Unfortunately, the data are not sufficient for detailed subgroup analyses due to the small number of individuals in the study. We added this to the limitation section: Due to the small sample size, subgroup analyses, e. g. on the interindividual response to music therapy at a given gestational age but at different corresponding postnatal ages could not be carried out.”

COMMENT 2

The levels of vigilance are described twice in the methods - in both the intervention and statistical analysis sections.

We have deleted the vigilance level description in the methods section.

COMMENT 3

Table 1 has inconsistent use of square brackets vs parentheses (e.g. for range of antibiotic treatment in the intervention group) and dots vs commas to show fractional numbers (dots used for SDs of GS but commas used for average antibiotic days).

In Table 1 we have replaced the commas with dots and made sure there are only parentheses. Where applicable, we introduced a new line to reduce the number of brackets per row.

COMMENT 4

There appears to be a typo on page 3 lines 113-114 regarding the average music therapy session length. Some brackets or parentheses may be missing.

The mean duration of each music therapy session was 19.3 ± 4.3 minutes (range 10 and 50 minutes). The sign ± was was shown incorrectly. We corrected the error in the manuscript.

COMMENT 5

Heart rate could perhaps be presented in the units beats per minute rather than just /min.

We have replaced from /min. to beats per min and breaths per min, respectively.

COMMENT 6

For the large tables with lots of confidence intervals like Table 3, it would be useful to bold or somehow highlight the CIs that do not cross 1.

We assume that with “CIs that do not cross the 1” you refer to non-overlapping CIs, as we did not calculate ratios. We bolded non-overlapping confidence intervals within rows. If that was not meant, please let us know.

Reviewer 2 Report

I suggest to the authors that the introduction better detail the benefits of music therapy for premature babies. Improve the background by describing results such as the study by van Dokkum et al. (2020) to understand the need to replicate a similar study. The discussion describes very brief comparison with other studies. Why not use a validated instrument to quantify reactions to music therapy?  

Author Response

Dear Reviewer,

we highly appreciate your critical input and the opportunity to improve our manuscript entitled "Music therapy is effective during sleep in preterm infants". Our changes to the original manuscript are marked using the “Track Changes” function.

All comments were addressed in a point-to-point fashion below:

Comments from Peer-Reviewer:

I suggest to the authors that the introduction better detail the benefits of music therapy for premature babies. Improve the background by describing results such as the study by van Dokkum et al. (2020) to understand the need to replicate a similar study.

We added information on the results by van Dokkum et al. to the introduction and revised the beginning of the last paragraph. We hope that the rationale to conduct our study comes out more clearly now.

The discussion describes very brief comparison with other studies. Why not use a validated instrument to quantify reactions to music therapy?  

To our knowledge, there are no validated tools to assess the effect of interventions such as music therapy on the behaviour of preterm infants, therefore we used the documentation of vital signs as a substitute. Even though the COMFORT-Neo scale was designed for titrating optimum levels of sedation during mechanical ventilation, we will consider it for future documentation of behavioural states before and after therapy. Van Dokkum et al. showed that it can be useful in this context.